# Genetic changes and testing associated with childhood glaucoma: A systematic review

**Anika Kumar, Ying Han, Julius T. Oatts** [ID] *

Department of Ophthalmology, University of California San Francisco, San Francisco, California, United States of America

* julius.oatts@ucsf.edu

## Abstract

Many forms of childhood glaucoma have been associated with underlying genetic changes, and variants in many genes have been described. Currently, testing is variable as there are no widely accepted guidelines for testing. This systematic review aimed to summarize the literature describing genetic changes and testing practices in childhood glaucoma. This systematic review was conducted in accordance with the Preferred Reporting Items for Systematic review and Meta-Analyses (PRISMA) 2020 guidelines and registered with Prospero (ID CRD42023400467). A comprehensive review of Pubmed, Embase, and Cochrane databases was performed from inception through March 2, 2023 using the search terms: (glaucoma) AND (pediatric OR childhood OR congenital OR child OR infant OR infantile) AND (gene OR genetic OR genotype OR locus OR genomic OR mutation OR variant OR test OR screen OR panel). Information was extracted regarding genetic variants including genotype-phenotype correlation. Risk of bias was assessed using the Newcastle-Ottawa Scale. Of 1,916 records screened, 196 studies met inclusion criteria and 53 genes were discussed. Among study populations, mean age±SD at glaucoma diagnosis was 8.94±9.54 years and 50.4% were male. The most common gene discussed was *CYP1B1*, evaluated in 109 (55.6%) studies. *CYP1B1* variants were associated with region and population-specific prevalence ranging from 5% to 86% among those with primary congenital glaucoma. *MYOC* variants were discussed in 31 (15.8%) studies with prevalence up to 36% among patients with juvenile open angle glaucoma. *FOXC1* variants were discussed in 25 (12.8%) studies, which demonstrated phenotypic severity dependent on degree of gene expression and type of mutation. Overall risk of bias was low; the most common domains of bias were selection and comparability. Numerous genes and genetic changes have been associated with childhood glaucoma. Understanding the most common genes as well as potential genotype-phenotype correlation has the potential to improve diagnostic and prognostic outcomes for children with glaucoma.

## Introduction

Glaucoma in children is a rare but potentially visually devastating condition characterized by elevated intraocular pressure, optic nerve damage, and the potential to cause irreversible

---

**Data Availability Statement:** All relevant data are within the manuscript and its Supporting Information files.

**Funding:** JTO received funding from the National Eye Institute/National Institutes of Health (NEI/NIH

K23EY034893, NEI/NIH EY002162 Core Grant for Vision Research, https://www.nei.nih.gov/). The funders did not play any role in the study design, data collection and analysis, decision to publish, or preparation of the manuscript.

**Competing interests:** The authors have declared that no competing interests exist.

blindness if not diagnosed and treated in a timely manner [1]. Childhood glaucoma is typically diagnosed clinically on the basis of intraocular pressure elevation, signs of glaucomatous optic nerve damage, corneal changes, or visual field defects consistent with glaucomatous optic nerve damage [2]. In some cases, genetic testing can establish a molecular diagnosis as many forms of childhood glaucoma, including primary congenital glaucoma (PCG), juvenile open angle glaucoma (JOAG), and glaucoma associated with non-acquired ocular or systemic diseases, have been associated with underlying genetic changes [3]. Understanding these genetic changes has the potential to shed light on pathophysiologic mechanisms of disease, disease prognostication, and treatment implications.

Currently, various clinical practice guidelines recommend that children at high risk of developing glaucoma should undergo an eye examination to detect disease [4–6]. Even though many genes have been implicated in the childhood glaucoma [7–9], no current guidelines outline specific protocols for populations who may be genetically "at increased risk." Additionally, for children with a confirmed diagnosis of glaucoma, the frequency and type of genetic testing is variable. This may be driven by the relative nascency of childhood glaucoma genetics that has not yet resulted in enough centralized high quality evidence to influence standard clinical practice, or the fact that genetic testing associated with childhood glaucoma can be inconsistent or inconclusive [10, 11]. This study summarizes the current body of evidence evaluating genetic changes and testing associated with childhood glaucoma.

## Materials and methods

### Inclusion and exclusion criteria

Studies were included in the systematic review if (1) they were prospective or retrospective cohort studies, cross-sectional studies, case-control studies, case series, or case reports, and (2) they specifically discussed genetic changes or testing associated with primary congenital glaucoma, juvenile-onset open angle glaucoma, secondary glaucoma associated with congenital non-acquired ocular anomalies, or unspecified glaucoma with age of onset between 0–18 years. Articles were excluded if (1) they were review articles, letters, or abstract-only publications (2) they discussed genetic changes or testing related to syndromic glaucoma with systemic features, (3) they lacked a child-specific analysis or discussion, or (4) they were not available as full-text articles in English.

### Search strategy

To ensure a comprehensive review of the available literature, Pubmed, Embase, and Cochrane databases were all queried using the following search terms: (glaucoma) AND (pediatric OR childhood OR congenital OR child OR infant OR infantile) AND (gene OR genetic OR genotype OR locus OR genomic OR mutation OR variant OR test OR screen OR panel). Additionally, relevant citations from papers identified through these databases were manually identified. All relevant studies published on or before March 2, 2023 were included.

**Study selection and data collection.** After searching the databases, all titles and abstracts were screened by a single reviewer (AK) to exclude irrelevant studies. Full text review was subsequently conducted in accordance with the aforementioned inclusion and exclusion criteria. Data was then extracted for all studies that met criteria by a single reviewer (AK). Data extracted included year of study, study design, sample size, mean age and sex breakdown of study population, etiology of glaucoma included in study, genes or genetic tests studied, specific genetic changes identified, and any quantitative measures reported in the study, such as diagnostic yield, prevalence of genetic changes, and genotype-phenotype correlations. An

independent validation of both the screening and data extraction process on a random 20% sample was conducted by a second reviewer (JO).

## Risk of bias assessment

A risk of bias assessment was then performed independently using the Newcastle-Ottawa Scale tool for cohort and case-control studies [12], as well as modified instruments for cross-sectional studies [13] and case reports and series [14], by two investigators (AK and JO). Disagreements were adjudicated by a third party (YH).

## Data synthesis and analysis

Results across studies were summarized using Microsoft Excel Version 16.0 (Redmond, WA) to provide descriptive statistics, including means and standard deviations of study population sizes and ages. This study did not require review by the Institutional Review Board because no patient data were included. This systematic review was conducted in accordance with the Preferred Reporting Items for Systematic review and Meta-Analyses (PRISMA) 2020 updated guidelines for reporting systematic reviews [15]. PRISMA checklist in S1 Checklist, Additionally, the methodological protocol of the search was registered with Prospero in February 2023 (ID CRD42023400467). Protocol in S1 Protocol.

## Study characteristics

Systematic search of the Pubmed, Embase, and Cochrane databases resulted in the identification of 2,349 studies published as of March 2, 2023. Following the removal of duplicates, 1,916 articles remained (Fig 1). Following exclusion of 1,255 of those studies based on screening of abstracts and titles alone, the remaining 661 studies underwent full-text review. Of those, 305 were excluded on the basis of relevance, 70 were excluded for being non-pediatric studies, 72 were excluded on the basis of study design, and 18 were excluded for not being available as full-texts in English. Following this assessment for eligibility, 196 studies were eligible for inclusion in the systematic review. A complete spreadsheet containing all data fields extracted from included studies can be found in S1 Appendix.

Of the 196 included studies, 36 (18.4%) were case reports, 40 (20.4%) were case series, 15 (7.7%) were case-control studies, 91 (66.9%) were cross-sectional studies, 7 (3.6%) were prospective cohort studies, and 7 (3.6%) were retrospective cohort studies. Twenty-three of the studies (11.7%) were published between 1993 and 2002, 58 (29.6%) between 2003 and 2012, and 115 (58.7%) between 2013 and March 2023. Within the studies, 53 unique genes were discussed. The most common gene discussed in the studies was CYP1B1, evaluated in 109 (55.6%) of the studies, followed by MYOC in 31 (15.8%), and FOXC1 in 25 (12.8%). Of the included studies that were not case reports, mean±SD number of study participants was 80.3 ±139.4 participants. The total number of participants included across all 196 studies was 12,607. Of the studies that published data on participant age, mean age±SD at glaucoma diagnosis was 8.94±9.54 years. Of the studies that published data on participant sex, an average of 50.4% were male. A comprehensive list of all genes, proposed functions, and glaucoma associations is shown in Table 1.

Overall, risk of bias was low among included studies. Of the case reports and series, 80% scored a 4 or higher out of 5 on the modified Newcastle-Ottawa scale for case series and reports, with the most common domain for bias being selection. Of the case-control studies, 70% scored a 6 or higher out of 9 on the Newcastle-Ottawa scale for case-control studies, with the most common domain for bias being comparability between cases and controls. Of the cross-sectional studies, 70% scored an 8 or higher out of 10 on the modified Newcastle-Ottawa

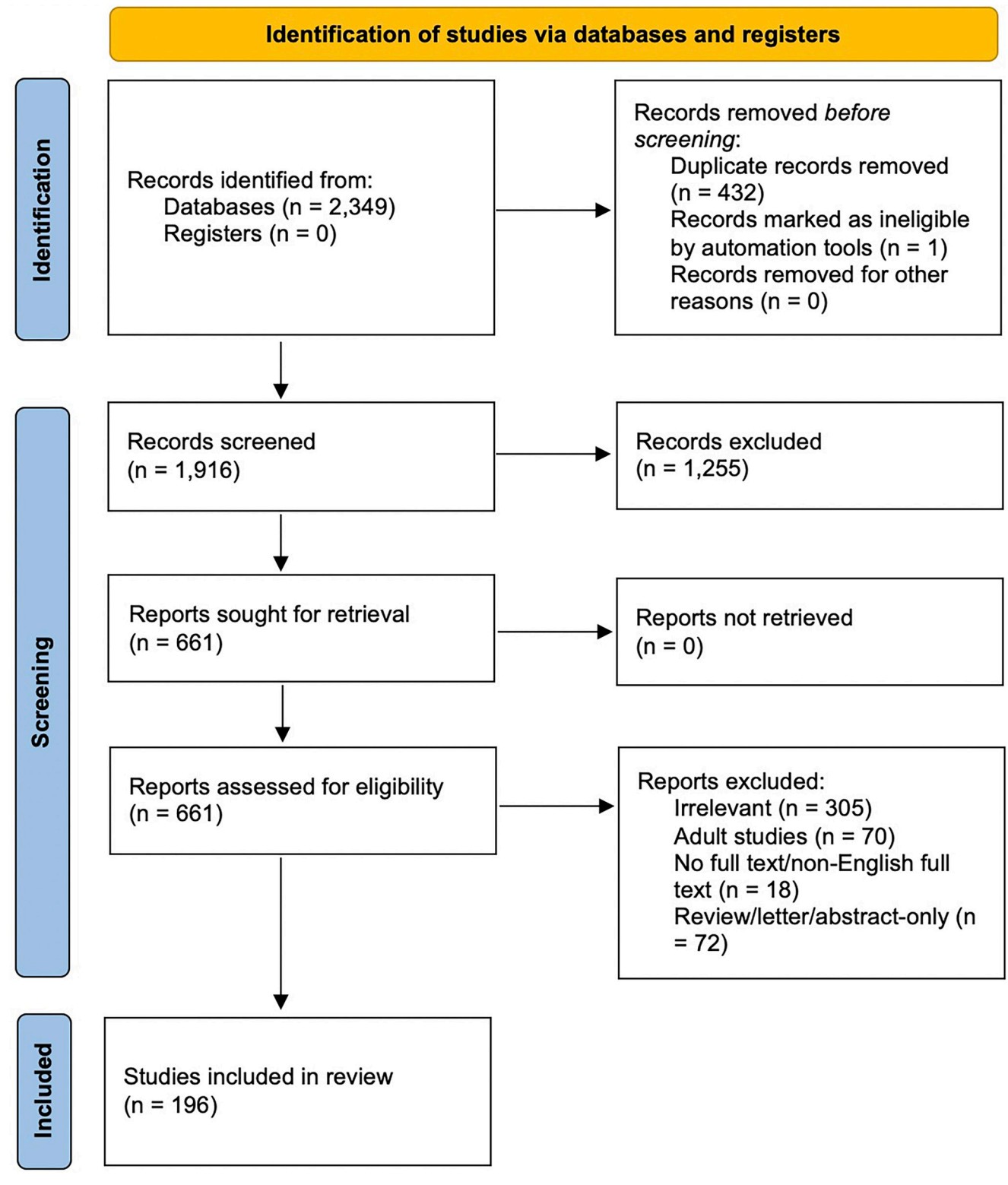

**Fig 1. PRISMA flow diagram.**

**Table 1. Nuclear genes associated with childhood glaucoma in published literature.**

| Gene | Protein | Relevant proposed function/expression of gene | Glaucoma types associated with gene |
|---|---|---|---|
| ADAM9 | a disintegrin and metalloprotease metallopeptidase domain 9 | Involved in cell-cell and cell-matrix interactions involved in neurogenesis | PCG [16] |
| ARX | aristaless related homeobox | Involved in central nervous system development | PCG [17] |
| ANGPT1 | angiopoietin 1 | Mediates matrix-endothelium interactions and is involved in vascular development | PCG, JOAG [18] |
| BEST1 | bestrophin 1 | Regulates ion transport in the retina | Angle-closure [19] |
| CHRDL1 | chordin like 1 | Regulates retinal angiogenesis in response to hypoxia | PCG [20] |
| COL1A1, COL18A1, COL2A1 | collagen type I alpha 1 chain, collagen type XVIII alpha 1 chain, collagen type II alpha 1 chain | Encodes fibrillar collagen found in cartilage and vitreous humor or eye | PCG [18], JOAG [18] |
| CPAMD8 | C3 and PZP like alpha-2-macroglobulin domain containing 8 | Involved in innate immunity and damage control | PCG, JOAG [21–23], glaucoma associated with non-acquired ocular anomalies [18] |
| CRYBB3 | crystallin beta B3 | Involved in maintaining the vertebrate eye lens | PCG [24], JOAG, glaucoma associated with non-acquired ocular anomalies [25] |
| CYP1B1 | cytochrome P450 family 1 subfamily B member 1 | Involved in metabolizing a signaling molecule involved in eye development | PCG [18, 20, 24, 26–121], JOAG [9, 122–127], glaucoma associated with non-acquired ocular anomalies [25, 128] |
| DPT | dermatopontin | Involved in extracellular matrix formation and cell-matrix interactions | PCG [129] |
| EFEMP1 | EGF containing fibulin extracellular matrix protein 1 | Encodes extracellular matrix glycoprotein involved in retinal drusen formation | JOAG [130] |
| FBN1 | fibrillin 1 | Encodes extracellular matrix protein expressed in the eye | PCG [131] |
| FOXC1 | forkhead box C1 | Regulates embryonic and ocular development and ocular drainage | PCG [24, 48, 72, 117, 129, 132–146], JOAG [18, 25, 128, 147], glaucoma associated with non-acquired ocular anomalies [148] |
| FYCO1 | FYVE and coiled-coil domain autophagy adaptor 1 | Mediates autophagy and expressed in the lens and retina | PCG [24] |
| GJA1, GJA8 | gap junction protein alpha 1, gap junction protein alpha 8 | Encodes connexin protein necessary for lens fiber growth and maturation | PCG [24, 117, 149], glaucoma associated with non-acquired ocular anomalies [25] |
| HMX1 | H6 family homeobox 1 | Involved in development of craniofacial structures | PCG [131] |
| LMX1B | LIM homeobox transcription factor 1 beta | Involved in development of the anterior segment of the eye | PCG [131] |
| LTBP2 | latent transforming growth factor beta binding protein 2 | Involved in ciliary microfibril development and lens suspension | PCG [24, 44, 68, 71, 82, 150–154], JOAG [123, 155], glaucoma associated with non-acquired ocular anomalies [25] |
| MAF | MAF bZIP transcription factor | Regulates embryonic lens fiber cell development | PCG [131] |
| MYOC | Myocilin, or trabecular meshwork glucocorticoid-inducible response (TIGR) | Involved in IOP regulation and expressed in ocular tissue | PCG [53–55, 68, 72, 74–76, 90, 156–159], JOAG [9, 118, 122, 123, 155, 160–172], glaucoma associated with non-acquired ocular anomalies [18] |
| OAT | ornithine aminotransferase | Involved in glutamate and GABA synthesis | Glaucoma associated with non-acquired ocular anomalies [173] |
| OPA1 | Optic atrophy type 1 mitochondrial dynamin like GTPase | Involved in mitochondrial metabolism in retinal ganglion cells | PCG, JOAG [166] |
| OPTN | Optineurin | Regulates basic cellular functions within trabecular meshwork and retina | PCG [18, 174], JOAG [164] |
| NTF4 | neurotrophin 4 | Regulates survival and differentiation of mammalian neurons | PCG [90] |

*(Continued)*

**Table 1.** (Continued)

| Gene | Protein | Relevant proposed function/expression of gene | Glaucoma types associated with gene |
|------|---------|----------------------------------------------|-------------------------------------|
| PAX6 | paired box 6 | Provides transcriptional regulation of neural development, especially in the eye | PCG [175–181], JOAG [18, 182] |
| PITX2, PITX3 | paired like homeodomain 2, paired like homeodomain 3 | Regulates development of the anterior segment of the eye | PCG [17, 72, 117, 131, 140, 144, 183], JOAG [9, 147], glaucoma associated with non-acquired ocular anomalies [18, 148, 184] |
| PLOD2 | procollagen-lysine,2-oxoglutarate 5-dioxygenase 2 | Involved in membrane stability and expressed in the eye during embryogenesis | PCG [185] |
| PRDM5 | PR/SET domain 5 | Regulates fibrillar collagens in the eye | PCG [131] |
| PTBP2 | polypyrimidine tract binding protein 2 | Regulates neural development via repression of select adult protein isoforms until final maturation | PCG, JOAG [18] |
| PXDN | Peroxidasin | Involved in extracellular matrix formation and is expressed in the eye | PCG [150] |
| RAX | retina and anterior neural fold homeobox | Regulates retinal cell fate determination and ocular development | PCG [131] |
| SIX1, SIX6 | SIX homeobox 1, SIX homeobox 6 | Involved in ocular development | PCG [131] |
| SLC4A11 | solute carrier family 4 member 11 | Encodes ion channel expressed in corneal endothelium | PCG [131], JOAG [18] |
| SOX11 | SRY-box transcription factor 11 | Regulates embryonic development and cell fate determination of ocular structures | PCG [24, 131], glaucoma associated with non-acquired ocular anomalies [25] |
| SVEP1 | sushi, von Willebrand factor type A, EGF and pentraxin domain containing 1 | Involved in epidermis development and lymph vessel morphogenesis | PCG [186] |
| TBK1 | TANK binding kinase 1 | Regulates autophagy in the retinal ganglion cell layer | PCG, JOAG [18] |
| TEK | TEK receptor tyrosine kinase | mediates embryonic vascular development through angiopoietin signaling | PCG [24, 37, 71, 186, 187], JOAG [18], glaucoma associated with non-acquired ocular anomalies [25] |
| THBS1 | thrombospondin 1 | Mediates cell-cell and cell-matrix interactions on ocular tissue | PCG [188] |
| TMEM98 | transmembrane protein 98 | Expressed in ocular tissues and regulates eye size | PCG, JOAG [18] |
| TNF | tumor necrosis factor | Involved in multifunctional inflammatory cytokine pathway | PCG [189] |
| TRIM44 | tripartite motif containing 44 | Regulates differentiation and maturation of neuronal cells | PCG [131] |
| WDR36 | WD repeat domain 36 | Involved in ocular tissue cell cycle progression, signal transduction, apoptosis, and gene regulation | PCG [90], JOAG [166] |
| WT1 | WT1 transcription factor | Regulates progenitor proliferation and retinal ganglion cells during retinogenesis | PCG [131] |
| VAX1 | ventral anterior homeobox 1 | Regulates development and morphogenesis of anterior ventral forebrain and visual system | PCG [131] |

scale for cross-sectional studies, with the most common domain for bias being comparability between different outcome groups. Of the cohort studies, 100% scored a 6 or higher out of 9 on the Newcastle-Ottawa scale for cohort studies, with the most common domain of bias being selection.

## Limitations

This systematic review is limited in that reported summary estimates may have been subject to publication bias; it is possible that reported metrics, such as diagnostic yield or magnitude of

genotype-phenotype correlations, may be overestimates of true estimates due to the tendency for positive findings to be overrepresented in the literature. Additionally, this review only included studies that had full text available in the English language, which may have resulted in incomplete summarizations of genetic changes and prevalence estimates by omitting studies in other languages conducted in globally diverse patient populations. Finally, though a comprehensive search strategy was implemented, it is possible that some relevant studies were not included due to variations in terminology or our use of only three major databases.

## Discussion/Summary of evidence

### CYP1B1

The *CYP1B1* gene, which encodes a cytochrome P450 family protein and is highly expressed in the eye, is arguably one of the most investigated genomic regions in the setting of childhood glaucoma. In PCG, *CYP1B1* variants are thought to be related to impaired metabolism of retinol, which disrupts retinoic acid levels required for ocular development [43]. Its increased expression in fetal eyes as compared to adult eyes suggests its significance in the development of childhood glaucoma specifically [190]. Numerous case reports have highlighted the incidence of bilateral PCG in those with homozygous or compound heterozygous *CYP1B1* variants in individuals both with and without a family history of the disease, with the most common variants being p.G61E, p.R368H, pE229K, and p.R390H [26–36].

Of analytical studies investigating the prevalence of genetic changes associated with childhood glaucoma, *CYP1B1* variants are the most common with varying prevalence across regions and populations. For example, among patients with PCG, cross-sectional studies have found the prevalence of *CYP1B1* variants to range between 5% and 23% in South Africa [71], China [46, 62], America [68], Vietnam [54], Japan [69, 94, 104, 107], and Germany [24], while prevalences range from 30% to 55% in studies from India [79, 95], Turkey [72, 98], Portugal [57, 128], Morocco [40], Spain [118], and France [103]. Among PCG patients, *CYP1B1* variants appear to be most prevalent in some South Asian and Middle Eastern populations as prevalences have been found to range from 64% to 85.7% in studies from Pakistan [99], Iran [63], and Saudi Arabia [47, 113, 123]. Prevalence of specific variants has also been found to be region- and population-specific. For example, among PCG patients, the prevalence of the missense p.G61E variant was found to be 7.8% in a Moroccan population [114], 47.1% in an Iranian population [119], 50% in an Israeli Bedouin population [49], and 63% in a Saudi Arabian population [66]. Additionally, while the frequency of the missense p.E378K variant was only 6.67% in a Mexican population [50], it was 100% in a Slovak population of patients with PCG, indicating a potential founder effect [77]. Studies have also evaluated the prevalence of *CYP1B1* variants in patients with various ocular anomalies. For example, the prevalence was found to be as high as 91% in patients with buphthalmos, including ectropion uveae and partial aniridia [87], 92.3% in patients with ectropion uveae [100], and 0% in patients with the Axenfeld-Rieger anomaly [117]. Together, this research highlights the relative prevalence of *CYP1B1* variants among cases of childhood glaucoma and also suggests that testing for such variants has varying degrees of utility depending on the patient population and ocular manifestations.

In evaluating genotype-phenotype correlations, studies have shown that those with homozygous *CYP1B1* variants generally display more severe clinical phenotypes compared to those without. For example, variants in *CYP1B1* have been associated with earlier age of disease onset [57, 75, 106, 191], higher likelihood of developing bilateral disease [57, 64, 75], higher intraocular pressure [64, 106], and requirement of more medical and surgical interventions [58, 75, 113]. However, among patients with *CYP1B1* genetic changes, penetrance is not full,

and phenotypic severity has been found to be variable, suggesting the presence of some type of genetic modification through interaction with other genes [89]. Several studies have explored whether the type of *CYP1B1* variant affects the phenotype. For example, in West Siberia, variants in codon 444 were associated with the most severe phenotypes, suggesting that codon's role in structural stabilization of the resulting protein [115]. Additionally, null variants have been found to be associated with a need for greater number of surgeries and earlier age of disease onset [33, 67]. In another study, the percentage of PCG patients with "severe" phenotypes was 100% in those with frameshift variants, 80% with missense p.E229K variants, and 66.7% with missense p.G61E variants [52]. However, in some familial studies, phenotypes of different degrees of severity have been observed among patients with the exact same variant, even within the same family, demonstrating that variant alone cannot account for all phenotypic differences [45, 121]. Overall, future studies investigating the effect of *CYP1B1* variants in both functional protein models and human correlates will be essential in predicting disease course and phenotypic severity of those with variants.

## FOXC1 and PITX2

The *FOXC1* gene encodes the forkhead box C1 protein, which is a transcription factor foundational in the regulation of embryonic and ocular development and highly expressed in important ocular structures including the iris, cornea, and trabecular meshwork [192]. Many case reports have described a spectrum of conditions associated with variants in this gene, most commonly including the Axenfeld-Rieger anomaly, as well as aniridia and megalocornea in the setting of heterozygous *FOXC1* variants [132–136]. Frequently reported variants associated with varying degrees of phenotypic severity in case series include missense variants of the arginine residue at position 127 [129, 138, 146], deletions [48, 137, 142], and duplications [139]. Among cross-sectional studies in German, Australian, Italian, and Spanish populations of patients with PCG and glaucoma associated with non-acquired ocular anomalies, the prevalence of *FOXC1* variants appears to range between 4% and 7.5% [24, 25, 143–145]. Through functional protein analysis, it has been proposed that a dose-dependent relationship exists between *FOXC1* expression and phenotype where variants that result in 50–60% or 130–150% of transcriptional activity are associated with glaucoma, and activity beyond these levels result in more severe anterior segment anomalies and extraocular manifestations [141]. For example, in one study of Swiss families, it was found that those with duplications with hypothesized 150% transcriptional activity exhibited glaucoma with less phenotypic severity than those with a frameshift *FOXC1* variants that resulted in little to no transcriptional activity [128]. Overall, these studies demonstrate a significant amount of phenotypic heterogeneity associated with relatively prevalent changes in *FOXC1* and future research is required to delineate the hypomorphic and hypermorphic variants associated with the most severe phenotypes.

Of note, the *FOXC1* gene has significant functional interactions with the *PITX2* gene, another gene implicated in childhood glaucoma [193]. The *PITX2* gene encodes the paired-like homeodomain 2 protein, a transcription factor involved in negative regulation of the *FOXC1* gene. Loss of function *PITX2* variants result in inappropriately extensive activation of *FOXC1*-target genes [194]. Thus, variants in *PITX2* have been reported in glaucoma associated with Axenfeld-Rieger syndrome even in the absence of *FOXC1* variants [17, 184]. Though *FOXC1* and *PITX2* variants are thought to cause childhood glaucoma through a similar mechanism, studies have shown that *FOXC1* variants (as compared to *PITX2* variants) have significantly greater disease penetrance and earlier age of onset [147, 148]. However, one study observed that despite increased prevalence of disease at age 10 in those with *FOXC1* variants as compared to *PITX2* variants, difference in prevalence was no longer significant at age 25 [140].

Additionally, FOXC1 variants are potentially more likely to be associated with corneal abnormalities and need for glaucoma surgery than *PITX2* variants [148]. Overall, these studies highlight that identification of causative genes in patients with Axenfeld-Rieger syndrome may have implications in anticipating phenotypic severity, disease progression, and surgical intervention requirements; future research is required to particularize these relationships with age.

## LTBP2

The *LTBP2* gene encodes the latent transforming growth factor beta binding protein 2, an extracellular matrix protein thought to be essential in ciliary microfibril development and the development of correct lens placement and suspension. It is located within 1.5 Mb from the GLC3 locus, which has been linked to PCG in family linkage studies [7]. *LTBP2* variants have also been described in association with microspherophakia, megalocornea, and ectopia lentis: all non-acquired ocular anomalies that can co-exist with glaucoma. For example, reports have described compound heterozygous *LTBP2* variants and the coexistence of *LTBP2* variants in those with *MYOC* variants contributing to severe childhood glaucomatous phenotypes [155, 195]. Additionally, some familial observational case series have described missense and frameshift variants in Iranian and Pakistani pedigrees, noting that consanguinity was present in all studied families [150–152]. The prevalence of *LTBP2* variants in childhood glaucoma patients is population-specific. For example, no variants to date have been identified in cross-sectional studies of PCG and JOAG populations from China, South Africa, Saudi Arabia, or the United States [68, 71, 123, 153]. However, *LTBP2* variants have been identified in 4–5.6% of study participants with childhood glaucoma in Germany [24, 25] and 12.5% in India [44]. Additionally, a single p.R299X variant has been identified in 40.5% of patients with PCG that all originated from the Roma founder population, with homozygotes for the variant presenting with more severe ocular phenotypes than heterozygotes [82]. Collectively, these findings suggest that the prevalence of causal *LTBP2* variants may be region-specific, and that using *LTBP2* sequencing for molecular diagnosis may not be productive in certain populations. Future research examining the association between *LTBP2* variant prevalence and consanguinity in a variety of different locations will help elucidate populations in which *LTBP2* testing may be the most valuable.

## MYOC

The *MYOC* gene encodes the myocilin protein, also known as the trabecular meshwork glucocorticoid-inducible response (TIGR) protein, which in the eye is expressed primarily in trabecular meshwork tissue and thought to be an important contributor to the regulation of intraocular pressure [196]. Homozygous and heterozygous missense *MYOC* variants have been implicated in case reports and cases series of bilateral PCG and JOAG [156, 160] Some common variants identified include the missense p.P370L [163, 168, 169] and p.Q48H [157, 159] variants. Of note, the missense p.Q48H variant is thought to contribute to a consequential proportion of cases in India, with that variant alone found in 2.5% of PCG cases in an observational study in India [158]. In cross-sectional analyses, the prevalence of *MYOC* variants has been found to range between 2.3% and 2.6% in Chinese and Indian populations with PCG [53, 166]. The prevalence in patients with JOAG is higher and has been found to range between 4% and 36% among Iranian, Canadian, Spanish, American, and Chinese populations [9, 118, 122, 161, 166]. Additionally, a study of the age-based prevalence of *MYOC* variants found that *MYOC* variants were identified in 36% of American glaucomatous probands with juvenile-onset disease as compared to only 4% of probands with adult-onset disease [161]. Together, these studies demonstrate that screening for *MYOC* variants is of highest utility in patients with JOAG or members of families with history of early-onset glaucoma.

*MYOC* variants have also been found to have significant interactions with other genes implicated in childhood glaucoma. For example, one study found that patients with coexisting *MYOC* and *OPTN* variants had more severe ocular phenotypes than those with *MYOC* variants alone [164]. The *OPTN* gene codes for the optineurin protein, which is expressed during early stages of eye development and helps regulate cellular functions such as protein trafficking and NF-κB pathway maintenance in the trabecular meshwork and retina. Though this phenomenon has not been extensively characterized in humans, cellular studies have noted that *OPTN* upregulation results in increased stability of *MYOC* mRNA; thus, loss of function variants at the *OPTN* gene drive dysregulation of *MYOC* expression [197], providing a possible pathophysiological mechanism of their interaction. Another study found that those with concurrent *MYOC* and *CYP1B1* variants had a much earlier age of onset of disease than those with *MYOC* variants alone [9]. One hypothesis for this interaction is that the CYP1B1 protein may be involved in metabolism of endogenous steroids, which are known to induce the myocilin protein; thus metabolic derangements from *CYP1B1* variants may further exacerbate the ramifications of any mutant myocilin proteins [198, 199]. Overall, the role of multiple genes in potential modification of *MYOC* gene expression implies a common interaction pathway. Further studies of functional protein interactions and their resulting clinical manifestations will be useful in understanding the mechanisms by which *MYOC* variants contribute to glaucoma and which patients may be at risk for developing the most severe phenotypes.

## TEK

The *TEK* gene encodes the tunica interna endothelial cell kinase, which is a tyrosine kinase protein that mediates embryonic vascular development through angiopoietin signaling [37]. Though its exact function in the development of glaucoma remains unknown, *TEK* variants are thought to impair aqueous humor outflow and Schlemm's canal development [200]. Though no specific variants appear to be predominant among *TEK* variants described in the literature, estimates of the prevalence of *TEK* variants in general range from 4% to 5.9% among German, Chinese, Australian, and South African populations with PCG, JOAG, and glaucoma associated with non-acquired ocular anomalies [18, 24, 25, 71, 187]. Unlike other genetic changes associated with childhood glaucoma, studies have demonstrated that the phenotypic penetrance of *TEK* variants is relatively low. For example, in one study of *TEK* variants in Australian patients with early-onset glaucoma, only 75% of those with *TEK* variants exhibited bilateral glaucoma, as compared to at least 97% of those with *CYP1B1*, *LTBP2*, and *MYOC* variants, for example [18]. In another study of PCG in Chinese patients, penetrance was only 68.5% [187]. It is worth mentioning that these studies were limited in that they did not investigate the association between penetrance and type of variant in large sample sizes; thus, it is possible that a dose-dependent or protein-structure effect, or other relationship, exists between *TEK* gene expression and phenotype that has yet to be identified. Regardless, one possible explanation is that *TEK* gene expression is highly susceptible to the influence of interaction with other genes. For example, one study has proposed that the *SVEP1* gene could be a potent genetic modifier as *SVEP1* loss of function alleles were demonstrated to reduce TEK expression in vascular endothelial cells in animal models and correlate with increased disease severity in human families with PCG [186]. The *SVEP1* gene encodes an extracellular matrix glycoprotein involved in epidermal and lymph vessel development. Another study identified the coexistence of heterozygous *TEK* and *CYP1B1* variants in cases of PCG; they then conducted functional analyses demonstrating that recombinant *CYP1B1* proteins interacted with recombinant *TEK* proteins to decrease *TEK* signaling [37]. Further studies evaluating modulators of this gene's expressivity can help elucidate the pathophysiological mechanism by which it drives

glaucoma and help predict which patients with *TEK* variants may be at greatest risk of severe disease.

## Real world genetic testing practice

Though no standard guidelines exist regarding genetic testing for childhood glaucoma, several studies have investigated its use in the real world. For example, in a cross-sectional study of pediatric referral practices in India, patients with glaucoma and objective features suggesting an underlying genetic abnormality were less than half as likely to be referred for formal genetic evaluation when they met with ophthalmologists than when they met with geneticists [201]. Though these findings may not be generalizable to all provider practices, it suggests that in general, there is room to improve initiating genetic testing. One potential explanation for this may relate to providers' hesitation around the utility of testing relative to the potential financial and logistical expenditures. A study investigating the diagnostic yield of genetic testing of early-onset glaucoma patients in a real world practice setting found that next generation sequencing was able to identify a causative variant in only 19% of those tested [202]. Notably, diagnostic yield was 32% in patients with glaucoma onset before 3 years of age but only 5% in patients with onset after three years of age, suggesting more limited utility of testing for later onset glaucoma. Additionally, in a study of 39 patients with PCG referred to a pediatric ocular genetics service in England, diagnostic yield of whole exome sequencing was only 12.8% [203]. In another study of 28 preschool-aged probands with anterior segment dysgenesis, including glaucoma, diagnostic yield was 39%. Additionally, it was found that establishing a molecular diagnosis altered management in 18% of those patients through avoidance of additional unnecessary tests and initiation of surveillance for other extraocular manifestations [204]. The lack of consistent recommendations for genetic testing may also relate to other practical barriers to incorporation of genetics assessments into clinical practice, including shortages of qualified ophthalmic genetic counselors, which can result in long wait times for patients to be evaluated [205]. Overall, while existing research demonstrates promising data on the utility of real world genetic testing, especially in patients with earlier onset glaucoma, future research on its capability to inform disease management is necessary to help shape provider practice patterns.

## Conclusion

Numerous genes and genetic changes have been described in association with childhood glaucoma, with the most common being *CYP1B1*, *MYOC*, and *FOXC1*. There is significant variability in genotype-phenotype correlation based on the specific gene and variant identified. Studies of real world genetic testing reveal a relatively low diagnostic yield, which may limit the practicality of genetic testing with currently available tools. Understanding the underlying genetic changes associated with childhood glaucoma has the potential to improve diagnostic, prognostic, and potentially therapeutic outcomes for children with glaucoma.

## Supporting information

**S1 Checklist. Preferred Reporting Items for Systematic review and Meta-Analyses (PRISMA) checklist.**
(PDF)

**S1 Protocol. Prospero protocol.**
(PDF)

**S1 Appendix. Complete extracted characteristics from included studies.**
(XLSX)

## Author Contributions

**Conceptualization:** Anika Kumar, Julius T. Oatts.

**Data curation:** Anika Kumar, Julius T. Oatts.

**Formal analysis:** Anika Kumar.

**Investigation:** Anika Kumar, Ying Han, Julius T. Oatts.

**Methodology:** Anika Kumar, Julius T. Oatts.

**Supervision:** Julius T. Oatts.

**Writing – original draft:** Anika Kumar, Julius T. Oatts.

**Writing – review & editing:** Anika Kumar, Ying Han, Julius T. Oatts.

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
