## [Decision Letter · Decision Letter 0]

17 Jan 2024

PONE-D-23-41816Genetic changes and testing associated with childhood glaucoma: a systematic reviewPLOS ONE

Dear Dr. Oatts,

Thank you for submitting your manuscript to PLOS ONE. After careful consideration, we feel that it has merit but does not fully meet PLOS ONE’s publication criteria as it currently stands. Therefore, we invite you to submit a revised version of the manuscript that addresses the points raised during the review process.

There is a conflict in opinions between reviewers on whether all data underlying the manuscript is fully available or not. I have made an evaluation and found that all the data underlying the manuscript is available  without restriction as supporting information.

The  reviewer's  comment  "Data presentation in Table 1 should be improved upon. The author can include column for studies/references" is recommended to improve the manuscript but  it is not required  to make a decision for publication. As author, you  can address it if you can.

We look forward to receiving your revised manuscript.

Kind regards,

Petros C Cyrus Kayange

Academic Editor

PLOS ONE

Journal Requirements:

Reviewers' comments:

Reviewer's Responses to Questions

**Comments to the Author**

1. Is the manuscript technically sound, and do the data support the conclusions?

Reviewer #1: Yes

Reviewer #2: Yes

2. Has the statistical analysis been performed appropriately and rigorously? 

Reviewer #1: N/A

Reviewer #2: Yes

3. Have the authors made all data underlying the findings in their manuscript fully available?

Reviewer #1: No

Reviewer #2: Yes

4. Is the manuscript presented in an intelligible fashion and written in standard English?

Reviewer #1: Yes

Reviewer #2: Yes

5. Review Comments to the Author

Reviewer #1: In this study, Kumar et al have conducted a systematic review, summarizing the current body of evidence on the genetic changes and genetic testing associated with childhood glaucoma by reviewing 196 studies that met their inclusion criteria.

This is an important and relevant study that provides information on the various genetic variants associated with childhood glaucoma and insights to real world genetic testing practices. However I have some comments.

1. The authors should use line numbering to make it easier to evaluate the work

2. When was the literature search conducted and what time period was included in search of the database

3. The authors should include the data extraction form used for the study as well as the data that was analysed

4. Data presentation in Table 1 should be improved upon. The authors can include a column for studies/references

5. The total number of participants studied in the review have not been included. 

6. Authors have not mentioned the limitation of the study

Reviewer #2: 1. Page 12 of submission, Inclusion and exclusion criteria should read secondary glaucoma associated with ocular anomlies instead of primary glaucoma.

2. Page 19, sentence regarding reference 87 should mention that CYP1B1 positive patients were notable for partial aniridia and mild iris ectropion rather than a full blown phenotypic presentation of aniridia.

3. Page 23 last paragraph a brief explanation of OPTN is warranted as this is the first mention of the gene in the manuscript other than the table.

4. Page 25 same comment for SVEP1.

5. Page 25, last paragraph, the relative lack of geneticists and long wait time for genetic evaluation also plays a role in the lack of a consistent protocol for genetic testing for these patients along with the low yield and cost.

6. PLOS authors have the option to publish the peer review history of their article (what does this mean?). If published, this will include your full peer review and any attached files.

Reviewer #1: No

Reviewer #2: No

---

## [Author Response · Author response to Decision Letter 0]

24 Jan 2024

RESPONSE TO REVIEWERS

We thank the Editor and Reviewers for your thorough reviews and helpful comments on our manuscript, entitled, “Genetic changes and testing associated with childhood glaucoma: a systematic review” (PONE-D-23-41816). Your suggested revisions have strengthened our manuscript, and our responses to each comment are detailed below.

Editor

1. Comment: 1. Please ensure that your manuscript meets PLOS ONE's style requirements, including those for file naming. The PLOS ONE style templates can be found at https://journals.plos.org/plosone/s/file?id=wjVg/PLOSOne_formatting_sample_main_body.pdf and https://journals.plos.org/plosone/s/file?id=ba62/PLOSOne_formatting_sample_title_authors_affiliations.pdf.

Response: We thank the editor for these style requirement templates. We have incorporated a variety of formatting changes to adhere with all the style guidelines in the templates, including adding line numbering, re-formatting our title page, using the appropriate heading levels, and using brackets instead of parentheses for our in-text references. 

2. Comment: 2. We note that your Data Availability Statement is currently as follows: [All relevant data are within the manuscript and its Supporting Information files.]

Response: We confirm that our submission contains all the raw data required to replicate the results of our study.

3. Comment: Please include captions for your Supporting Information files at the end of your manuscript, and update any in-text citations to match accordingly. Please see our Supporting Information guidelines for more information: http://journals.plos.org/plosone/s/supporting-information.

Response: We have added captions for our 3 supporting files (S1 checklist, S2 protocol, and S3 appendix) at the end of our manuscript and appropriately updated the in-text citations (Line 122, 124, 135).

4. Comment: Please review your reference list to ensure that it is complete and correct. If you have cited papers that have been retracted, please include the rationale for doing so in the manuscript text, or remove these references and replace them with relevant current references. Any changes to the reference list should be mentioned in the rebuttal letter that accompanies your revised manuscript. If you need to cite a retracted article, indicate the article’s retracted status in the References list and also include a citation and full reference for the retraction notice.

Response: We have reviewed our reference list to ensure that it is complete, correct, and appropriately formatted.

Reviewer #1

1. Comment: In this study, Kumar et al have conducted a systematic review, summarizing the current body of evidence on the genetic changes and genetic testing associated with childhood glaucoma by reviewing 196 studies that met their inclusion criteria.

This is an important and relevant study that provides information on the various genetic variants associated with childhood glaucoma and insights to real world genetic testing practices. However I have some comments.

Response: We appreciate the reviewer comments that this article is an important and relevant study. We have addressed your points below.

2. Comment: 1. The authors should use line numbering to make it easier to evaluate the work.

Response: We thank the reviewer for this suggestion for facilitating evaluation of this work. We have included continuous line numbering in the manuscript file.

3. Comment: When was the literature search conducted and what time period was included in search of the database

Response: We appreciate this comment as an opportunity to clarify our literature search timeline. We included all studies that met our search criteria published any time on or before March 2, 2023. Currently, the paper includes the following statement in our search strategy subsection of our methods section: “All relevant studies published on or before March 2, 2023 were included” (Line 97). To further emphasize this point, we have modified the first statement of our study characteristics section to read “Systematic search of the Pubmed, Embase, and Cochrane databases resulted in the identification of 2,349 studies published as of March 2, 2023” (Line 129).

4. Comment: The authors should include the data extraction form used for the study as well as the data that was analysed.

Response: Thank you for this comment. As recommended by the editor, we have now included our data extraction form as well as all collected data for the 196 studies as a supporting file. A reference to this is now included in our study characteristics section that reads: “A complete spreadsheet containing all data fields extracted from included studies can be found in S3 Appendix.” (Line 135-136)

5. Comment: Data presentation in Table 1 should be improved upon. The authors can include a column for studies/references

Response: Thank you for this suggestion. While we understand the logic behind adding a column with each reference, given the number of references, we chose not to do this, as it would make our table significantly longer (currently 2.5 pages – would increase to 4 pages). As an alternative, we have included the relevant references for each row as superscripts and have also provided all data we collected as a supporting file. We hope these changes capture the spirit of this comment to improve the readability of Table 1 without increasing its length significantly.

6. Comment: The total number of participants studied in the review have not been included. 

Response: We thank the reviewer for this comment relating to the number of participants included in all studies. We have added the following line in our Study characteristics section to characterize the average study population size among studies included in the review and total number of individuals encompassed by the studies in the review: “Of the included studies that were not case reports, mean±SD number of study participants was 80.3±139.4 participants. The total number of participants included across all 196 studies was 12,607.,” (Line 144-147). 

7. Comment: Authors have not mentioned the limitation of the study

Response: We thank the reviewer for this opportunity to discuss the limitations of this review—while we alluded to some limitations throughout the text, we did not have an explicit limitations section in our manuscript. To address this, we have added a limitations sub-section in our Study characteristics section discussing publication bias, our use of only studies published in English, and limitations of our search strategy. (Line 162-171). 

Reviewer #2

1. Comment: Page 12 of submission, Inclusion and exclusion criteria should read secondary glaucoma associated with ocular anomlies instead of primary glaucoma.

Response: We thank the Reviewer for this clarification of terminology and agree with this change. We have modified our inclusion criteria to now read, “secondary glaucoma associated with congenital non-acquired ocular anomalies.” (Line 85)

2. Comment: Page 19, sentence regarding reference 87 should mention that CYP1B1 positive patients were notable for partial aniridia and mild iris ectropion rather than a full blown phenotypic presentation of aniridia.

Response: We thank the reviewer for highlighting this nuance related to the study population corresponding to reference 87. We have modified our description of the results from that study to more accurately reflect the patients evaluated. The statement now reads, “For example, the prevalence was found to be as high as 91% in patients with buphthalmos, including ectropion uveae and partial aniridia[87],” (Line 202). 

3. Comment: Page 23 last paragraph a brief explanation of OPTN is warranted as this is the first mention of the gene in the manuscript other than the table.

Response: We thank the reviewer for highlighting an opportunity to strengthen our description of the OPTN gene and agree that a short explanation of the gene would be helpful in this section. We have included the following statement in our MYOC subsection: “The OPTN gene codes for the optineurin protein, which is expressed during early stages of eye development and helps regulate cellular functions such as protein trafficking and NF-κB pathway maintenance in the trabecular meshwork and retina,” (Line 313-315).

4. Comment: Page 25 same comment for SVEP1.

Response: Similar to the previous comment, we agree a short description of the SVEP1 gene would be helpful in the TEK subsection. We included the following statement, “The SVEP1 gene encodes an extracellular matrix glycoprotein involved in epidermal and lymph vessel development,” (Line 352-353).

5. Comment: Page 25, last paragraph, the relative lack of geneticists and long wait time for genetic evaluation also plays a role in the lack of a consistent protocol for genetic testing for these patients along with the low yield and cost.

Response: We thank the reviewer for this important comment regarding potential explanations for the lack of consistent genetic testing recommendations, and appreciate the opportunity to additionally include discussion of practical barriers to implementation of widespread genetic testing. To comment on these issues, we have added the following statement and an accompanying reference to our real-world genetic testing section: “The lack of consistent recommendations for genetic testing may also relate to other practical barriers to incorporation of genetics assessments into clinical practice, including shortages of qualified ophthalmic genetic counselors, which can result in long wait times for patients to be evaluated.[206]” (Line 379-382).

---

## [Editor Report · Decision Letter 1]

1 Feb 2024

Genetic changes and testing associated with childhood glaucoma: a systematic review

PONE-D-23-41816R1

Dear Dr. Oatts,

We’re pleased to inform you that your manuscript has been judged scientifically suitable for publication and will be formally accepted for publication once it meets all outstanding technical requirements.

Kind regards,

Petros  Cyrus Kayange

Academic Editor

PLOS ONE
---

## [Editor Report · Acceptance letter]

13 Feb 2024

PONE-D-23-41816R1 

PLOS ONE

Dear Dr. Oatts, 

I'm pleased to inform you that your manuscript has been deemed suitable for publication in PLOS ONE. Congratulations! Your manuscript is now being handed over to our production team.

Kind regards, 

on behalf of

Dr. Petros C Cyrus Kayange 

Academic Editor

PLOS ONE